# A Framework for Individualizing Predictions of Disease Trajectories by Exploiting Multi-Resolution Structure

**Peter Schulam**
Dept. of Computer Science
Johns Hopkins University
Baltimore, MD 21218
pschulam@jhu.edu

**Suchi Saria**
Dept. of Computer Science
Johns Hopkins University
Baltimore, MD 21218
ssaria@cs.jhu.edu

## Abstract

For many complex diseases, there is a wide variety of ways in which an individual can manifest the disease. The challenge of personalized medicine is to develop tools that can accurately predict the trajectory of an individual's disease, which can in turn enable clinicians to optimize treatments. We represent an individual's disease trajectory as a continuous-valued continuous-time function describing the severity of the disease over time. We propose a hierarchical latent variable model that individualizes predictions of disease trajectories. This model shares statistical strength across observations at different resolutions–the population, subpopulation and the individual level. We describe an algorithm for learning population and subpopulation parameters offline, and an online procedure for dynamically learning individual-specific parameters. Finally, we validate our model on the task of predicting the course of interstitial lung disease, a leading cause of death among patients with the autoimmune disease scleroderma. We compare our approach against state-of-the-art and demonstrate significant improvements in predictive accuracy.

## 1 Introduction

In complex, chronic diseases such as autism, lupus, and Parkinson's, the way the disease manifests may vary greatly across individuals [1]. For example, in scleroderma, the disease we use as a running example in this work, individuals may be affected across six organ systems—the lungs, heart, skin, gastrointestinal tract, kidneys, and vasculature—to varying extents [2]. For any single organ system, some individuals may show rapid decline throughout the course of their disease, while others may show early decline but stabilize later on. Often in such diseases, the most effective drugs have strong side-effects. With tools that can accurately predict an individual's *disease activity trajectory*, clinicians can more aggressively treat those at greatest risk early, rather than waiting until the disease progresses to a high level of severity. To monitor the disease, physicians use clinical markers to quantify severity. In scleroderma, for example, PFVC is a clinical marker used to measure lung severity. The task of individualized prediction of disease activity trajectories is that of using an individual's clinical history to predict the future course of a clinical marker; in other words, the goal is to predict a *function* representing a trajectory that is updated *dynamically* using an individual's previous markers and individual characteristics.

Predicting disease activity trajectories presents a number of challenges. First, there are multiple *latent factors* that cause heterogeneity across individuals. One such factor is the underlying biological mechanism driving the disease. For example, two different genetic mutations may trigger distinct disease trajectories (e.g. as in Figures 1a and 1b). If we could divide individuals into groups according to their mechanisms—or disease *subtypes* (see e.g. [3, 4, 5, 6])—it would be straightforward to fit separate models to each *subpopulation*. In most complex diseases, however, the mechanisms are poorly understood and clear definitions of subtypes do not exist. If subtype alone determined trajectory, then we could cluster individuals. However, other unobserved *individual-specific* factors

such as behavior and prior exposures affect health and can cause different trajectories across individuals of the same subtype. For instance, a chronic smoker will typically have unhealthy lungs and so may have a trajectory that is consistently lower than a non-smoker's, which we must account for using individual-specific parameters. An individual's trajectory may also be influenced by *transient factors*—e.g. an infection unrelated to the disease that makes it difficult to breath (similar to the "dips" in Figure 1c or the third row in Figure 1d). This can cause marker values to temporarily drop, and may be hard to distinguish from disease activity. We show that these factors can be arranged in a hierarchy (population, subpopulation, and individual), but that not all levels of the hierarchy are observed. Finally, the functional outcome is a rich target, and therefore more challenging to model than scalar outcomes. In addition, the marker data is observed in continuous-time and is irregularly sampled, making commonly used discrete-time approaches to time series modeling (or approaches that rely on imputation) not well suited to this domain.

**Related work.** The majority of predictive models in medicine explain variability in the target outcome by conditioning on observed risk factors alone. However, these do not account for latent sources of variability such as those discussed above. Further, these models are typically cross-sectional—they use features from data measured up until the current time to predict a clinical marker or outcome at a fixed point in the future. As an example, consider the mortality prediction model by Lee et al. [7], where logistic regression is used to integrate features into a prediction about the probability of death within 30 days for a given patient. To predict the outcome at multiple time points, typically separate models are trained. Moreover, these models use data from a fixed-size window, rather than a growing history.

Researchers in the statistics and machine learning communities have proposed solutions that address a number of these limitations. Most related to our work is that by Rizopoulos [8], where the focus is on making dynamical predictions about a time-to-event outcome (e.g. time until death). Their model updates predictions over time using all previously observed values of a longitudinally recorded marker. Besides conditioning on observed factors, they account for latent heterogeneity across individuals by allowing for individual-specific adjustments to the population-level model—e.g. for a longitudinal marker, deviations from the population baseline are modeled using random effects by sampling individual-specific intercepts from a common distribution. Other closely related work by Proust-Lima et al. [9] tackle a similar problem as Rizopoulos, but address heterogeneity using a mixture model.

Another common approach to dynamical predictions is to use Markov models such as order-$p$ autoregressive models (AR-$p$), HMMs, state space models, and dynamic Bayesian networks (see e.g. in [10]). Although such models naturally make dynamic predictions using the full history by forward-filtering, they typically assume discrete, regularly-spaced observation times. Gaussian processes (GPs) are a commonly used alternative for handling continuous-time observations—see Roberts et al. [11] for a recent review of GP time series models. Since Gaussian processes are non-parametric generative models of functions, they naturally produce functional predictions dynamically by using the posterior predictive conditioned on the observed data. Mixtures of GPs have been applied to model heterogeneity in the covariance structure across time series (e.g. [12]), however as noted in Roberts et al., appropriate mean functions are critical for accurate forecasts using GPs. In our work, an individual's trajectory is expressed as a GP with a highly structured mean comprising population, subpopulation and individual-level components where some components are observed and others require inference.

More broadly, multi-level models have been applied in many fields to model heterogeneous collections of units that are organized within a hierarchy [13]. For example, in predicting student grades over time, individuals within a school may have parameters sampled from the school-level model, and the school-level model parameters in turn may be sampled from a county-specific model. In our setting, the hierarchical structure—which individuals belong to the same subgroup—is not known *a priori*. Similar ideas are studied in multi-task learning, where relationships between distinct prediction tasks are used to encourage similar parameters. This has been applied to modeling trajectories by treating predictions at each time point as a separate task and enforcing similarity between submodels close in time [14]. This approach is limited, however, in that it models a finite number of times. Others, more recently, have developed models for disease trajectories (see [15, 16] and references within) but these focus on retrospective analysis to discover disease etiology rather than dynamical prediction. Schulam et al. [16] incorporate differences in trajectories due to subtypes and individual-specific factors. We build upon this work here. Finally, recommender systems also share

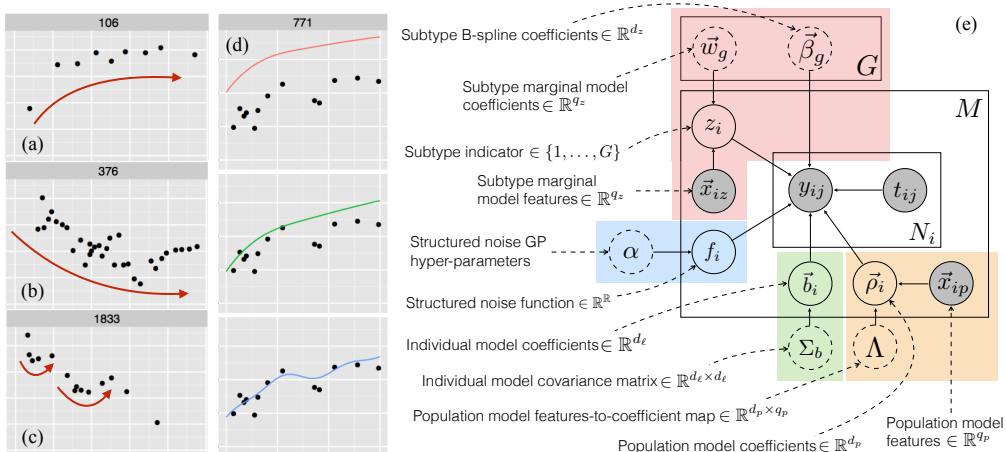

Figure 1: Plots (a-c) show example marker trajectories. Plot (d) shows adjustments to a population and subpopulation fit (row 1). Row 2 makes an individual-specific long-term adjustment. Row 3 makes short-term structured noise adjustments. Plot (e) shows the proposed graphical model. Levels in the hierarchy are color-coded. Model parameters are enclosed in dashed circles. Observed random variables are shaded.

information across individuals with the aim of tailoring predictions (see e.g. [17, 18, 19]), but the task is otherwise distinct from ours.

**Contributions.** We propose a hierarchical model of disease activity trajectories that directly addresses common—latent and observed—sources of heterogeneity in complex, chronic diseases using three levels: the population level, subpopulation level, and individual level. The model discovers the subpopulation structure automatically, and infers individual-level structure over time when making predictions. In addition, we include a Gaussian process as a model of structured noise, which is designed to explain away temporary sources of variability that are unrelated to disease activity. Together, these four components allow individual trajectories to be highly heterogeneous while simultaneously sharing statistical strength across observations at different "resolutions" of the data. When making predictions for a given individual, we use Bayesian inference to dynamically update our posterior belief over individual-specific parameters given the clinical history and use the posterior predictive to produce a trajectory estimate. Finally, we evaluate our approach by developing a state-of-the-art trajectory prediction tool for lung disease in scleroderma. We train our model using a large, national dataset containing individuals with scleroderma tracked over 20 years and compare our predictions against alternative approaches. We find that our approach yields significant gains in predictive accuracy of disease activity trajectories.

## 2   Disease Trajectory Model

We describe a hierarchical model of an individual's clinical marker values. The graphical model is shown in Figure 1e. For each individual $i$, we use $N_i$ to denote the number of observed markers. We denote each individual observation using $y_{ij}$ and its measurement time using $t_{ij}$ where $j \in \{1, \dots, N_i\}$. We use $\vec{y}_i \in \mathbb{R}^{N_i}$ and $\vec{t}_i \in \mathbb{R}^{N_i}$ to denote all of individual $i$'s marker values and measurement times respectively. In the following discussion, $\Phi(t_{ij})$ denotes a column-vector containing a basis expansion of the time $t_{ij}$ and we use $\Phi\left(\vec{t}_i\right) = [\Phi(t_{i1}), \dots, \Phi(t_{iN_i})]^\top$ to denote the matrix containing the basis expansion of points in $\vec{t}_i$ in each of its rows. We model the $j$th marker value for individual $i$ as a normally distributed random variable with a mean assumed to be the sum of four terms: a population component, a subpopulation component, an individual component, and a structured noise component:

$$y_{ij} \sim \mathcal{N}\left( \underbrace{\Phi_p(t_{ij})^\top \Lambda \, \vec{x}_{ip}}_{\text{(A) population}} + \underbrace{\Phi_z(t_{ij})^\top \vec{\beta}_{z_i}}_{\text{(B) subpopulation}} + \underbrace{\Phi_\ell(t_{ij})^\top \vec{b}_i}_{\text{(C) individual}} + \underbrace{f_i(t_{ij})}_{\text{(D) structured noise}}, \sigma^2 \right). \tag{1}$$

The four terms in the sum serve two purposes. First, they allow for a number of different sources of variation to influence the observed marker value, which allows for heterogeneity both across and within individuals. Second, they share statistical strength across different subsets of observations. The population component shares strength across all observations. The subpopulation component

shares strength across observations belonging to subgroups of individuals. The individual component shares strength across all observations belonging to the same individual. Finally, the structured noise shares information across observations belonging to the same individual that are measured at similar times. Predicting an individual's trajectory involves estimating her subtype and individual-specific parameters as new clinical data becomes available[1]. We describe each of the components in detail below.

**Population level.** The population model predicts aspects of an individual's disease activity trajectory using *observed* baseline characteristics (e.g. gender and race), which are represented using the feature vector $\vec{x}_{ip}$. This sub-model is shown within the orange box in Figure 1e. Here we assume that this component is a linear model where the coefficients are a function of the features $\vec{x}_{ip} \in \mathbb{R}^{q_p}$. The predicted value of the $j$th marker of individual $i$ measured at time $t_{ij}$ is shown in Eq. 1 (A), where $\Phi_p(t) \in \mathbb{R}^{d_p}$ is a basis expansion of the observation time and $\Lambda \in \mathbb{R}^{d_p \times q_p}$ is a matrix used as a linear map from an individual's covariates $\vec{x}_{ip}$ to coefficients $\rho_i \in \mathbb{R}^{d_p}$. At this level, individuals with similar covariates will have similar coefficients. The matrix $\Lambda$ is learned offline.

**Subpopulation level.** We model an individual's subtype using a discrete-valued latent variable $z_i \in \{1, \ldots, G\}$, where $G$ is the number of subtypes. We associate each subtype with a unique disease activity trajectory represented using B-splines, where the number and location of the knots and the degree of the polynomial pieces are fixed prior to learning. These hyper-parameters determine a basis expansion $\Phi_z(t) \in \mathbb{R}^{d_z}$ mapping a time $t$ to the B-spline basis function values at that time. Trajectories for each subtype are parameterized by a vector of coefficients $\vec{\beta}_g \in \mathbb{R}^{d_z}$ for $g \in \{1, \ldots, G\}$, which are learned offline. Under subtype $z_i$, the predicted value of marker $y_{ij}$ measured at time $t_{ij}$ is shown in Eq. 1 (B). This component explains differences such as those observed between the trajectories in Figures 1a and 1b. In many cases, features at baseline may be predictive of subtype. For example, in scleroderma, the types of antibody an individual produces (i.e. the presence of certain proteins in the blood) are correlated with certain trajectories. We can improve predictive performance by conditioning on baseline covariates to infer the subtype. To do this, we use a multinomial logistic regression to define feature-dependent marginal probabilities: $z_i \mid \vec{x}_{iz} \sim \text{Mult}\left(\pi_{1:G}\left(\vec{x}_{iz}\right)\right)$, where $\pi_g\left(\vec{x}_{iz}\right) \propto e^{\vec{w}_g^\top \vec{x}_{iz}}$. We denote the weights of the multinomial regression using $\vec{w}_{1:G}$, where the weights of the first class are constrained to be $\vec{0}$ to ensure model identifiability. The remaining weights are learned offline.

**Individual level.** This level models deviations from the population and subpopulation models using parameters that are learned dynamically as the individual's clinical history grows. Here, we parameterize the individual component using a linear model with basis expansion $\Phi_\ell(t) \in \mathbb{R}^{d_\ell}$ and individual-specific coefficients $\vec{b}_i \in \mathbb{R}^{d_\ell}$. An individual's coefficients are modeled as latent variables with marginal distribution $\vec{b}_i \sim \mathcal{N}(\vec{0}, \Sigma_b)$. For individual $i$, the predicted value of marker $y_{ij}$ measured at time $t_{ij}$ is shown in Eq. 1 (C). This component can explain, for example, differences in overall health due to an unobserved characteristic such as chronic smoking, which may cause atypically lower lung function than what is predicted by the population and subpopulation components. Such an adjustment is illustrated across the first and second rows of Figure 1d.

**Structured noise.** Finally, the structured noise component $f_i$ captures transient trends. For example, an infection may cause an individual's lung function to temporarily appear more restricted than it actually is, which may cause short-term trends like those shown in Figure 1c and the third row of Figure 1d. We treat $f_i$ as a function-valued latent variable and model it using a Gaussian process with zero-valued mean function and Ornstein-Uhlenbeck (OU) covariance function: $K_{\text{OU}}(t_1, t_2) = a^2 \exp\left\{-\ell^{-1}|t_1 - t_2|\right\}$. The amplitude $a$ controls the magnitude of the structured noise that we expect to see and the length-scale $\ell$ controls the length of time over which we expect these temporary trends to occur. The OU kernel is ideal for modeling such deviations as it is both mean-reverting and draws from the corresponding stochastic process are only first-order continuous, which eliminates long-range dependencies between deviations [20]. Applications in other domains may require different kernel structures motivated by properties of the noise in the trajectories.

## 2.1 Learning

**Objective function.** To learn the parameters of our model $\Theta = \{\Lambda, \vec{w}_{1:G}, \vec{\beta}_{1:G}, \Sigma_b, a, \ell, \sigma^2\}$, we maximize the observed-data log-likelihood (i.e. the probability of all individual's marker values $\vec{y}_i$ given measurement times $\vec{t}_i$ and features $\{\vec{x}_{ip}, \vec{x}_{iz}\}$). This requires marginalizing over the latent variables $\{z_i, \vec{b}_i, f_i\}$ for each individual. This yields a mixture of multivariate normals:

$$P\left(\vec{y}_i \mid X_i, \Theta\right) = \sum_{z_i=1}^{G} \pi_{z_i}\left(\vec{x}_{iz}\right) \mathcal{N}\left(\vec{y}_i \mid \Phi_p\left(\vec{t}_i\right) \Lambda \vec{x}_{ip} + \Phi_z\left(\vec{t}_i\right) \vec{\beta}_{z_i}, K\left(\vec{t}_i, \vec{t}_i\right)\right), \qquad (2)$$

where $K(t_1, t_2) = \Phi_\ell(t_1)^\top \Sigma_b \Phi_\ell(t_2) + K_{\text{OU}}(t_1, t_2) + \sigma^2 \mathbb{I}(t_1 = t_2)$. The observed-data log-likelihood for all individuals is therefore: $\mathcal{L}(\Theta) = \sum_{i=1}^{M} \log P\left(\vec{y}_i \mid X_i, \Theta\right)$. A more detailed derivation is provided in the supplement.

**Optimizing the objective.** To maximize the observed-data log-likelihood with respect to $\Theta$, we partition the parameters into two subsets. The first subset, $\Theta_1 = \{\Sigma_b, \alpha, \ell, \sigma^2\}$, contains values that parameterize the covariance function $K(t_1, t_2)$ above. As is often done when designing the kernel of a Gaussian process, we use a combination of domain knowledge to choose candidate values and model selection using observed-data log-likelihood as a criterion for choosing among candidates [20]. The second subset, $\Theta_2 = \{\Lambda, \vec{w}_{1:G}, \vec{\beta}_{1:G}\}$, contains values that parameterize the mean of the multivariate normal distribution in Equation 2. We learn these parameters using expectation maximization (EM) to find a local maximum of the observed-data log-likelihood.

**Expectation step.** All parameters related to $\vec{b}_i$ and $f_i$ are limited to the covariance kernel and are not optimized using EM. We therefore only need to consider the subtype indicators $z_i$ as unobserved in the expectation step. Because $z_i$ is discrete, its posterior is computed by normalizing the joint probability of $z_i$ and $\vec{y}_i$. Let $\pi_{ig}^*$ denote the posterior probability that individual $i$ has subtype $g \in \{1, \ldots, G\}$, then we have

$$\pi_{ig}^* \propto \pi_g\left(\vec{x}_{iz}\right) \mathcal{N}\left(\vec{y}_i \mid \Phi_p\left(\vec{t}_i\right) \Lambda \vec{x}_{ip} + \Phi_z\left(\vec{t}_i\right) \vec{\beta}_g, K\left(\vec{t}_i, \vec{t}_i\right)\right). \qquad (3)$$

**Maximization step.** In the maximization step, we optimize the marginal probability of the soft assignments under the multinomial logistic regression model with respect to $\vec{w}_{1:G}$ using gradient-based methods. To optimize the expected complete-data log-likelihood with respect to $\Lambda$ and $\vec{\beta}_{1:G}$, we note that the mean of the multivariate normal for each individual is a linear function of these parameters. Holding $\Lambda$ fixed, we can therefore solve for $\vec{\beta}_{1:G}$ in closed form and vice versa. We use a block coordinate ascent approach, alternating between solving for $\Lambda$ and $\vec{\beta}_{1:G}$ until convergence. Because the expected complete-data log-likelihood is concave with respect to all parameters in $\Theta_2$, each maximization step is guaranteed to converge. We provide additional details in the supplement.

## 2.2 Prediction

Our prediction $\hat{y}(t_i')$ for the value of the trajectory at time $t_i'$ is the expectation of the marker $y_i'$ under the posterior predictive conditioned on observed markers $\vec{y}_i$ measured at times $\vec{t}_i$ thus far. This requires evaluating the following expression:

$$\hat{y}\left(t_i'\right) = \sum_{z_i=1}^{G} \int_{R^{d_\ell}} \int_{R^{N_i}} \underbrace{\mathbb{E}\left[y_i' \mid z_i, \vec{b}_i, f_i, t_i'\right]}_{\text{prediction given latent vars.}} \underbrace{P\left(z_i, \vec{b}_i, f_i \mid \vec{y}_i, X_i, \Theta\right)}_{\text{posterior over latent vars.}} df_i \, d\vec{b}_i \qquad (4)$$

$$= \mathbb{E}_{z_i, \vec{b}_i, f_i}^* \left[\Phi_p\left(t_i'\right)^\top \Lambda \vec{x}_{ip} + \Phi_z\left(t_i'\right)^\top \vec{\beta}_{z_i} + \Phi_\ell\left(t_i'\right)^\top \vec{b}_i + f_i\left(t_i'\right)\right] \qquad (5)$$

$$= \underbrace{\Phi_p\left(t_i'\right)^\top \Lambda \vec{x}_{ip}}_{\text{population prediction}} + \underbrace{\Phi_z\left(t_i'\right)^\top \overbrace{\mathbb{E}_{z_i}^*\left[\vec{\beta}_{z_i}\right]}^{\vec{\beta}_i^* \text{ (Eq. 7)}}}_{\text{subpopulation prediction}} + \underbrace{\Phi_\ell\left(t_i'\right)^\top \overbrace{\mathbb{E}_{\vec{b}_i}^*\left[\vec{b}_i\right]}^{\vec{b}_i^* \text{ (Eq. 10)}}}_{\text{individual prediction}} + \underbrace{\overbrace{\mathbb{E}_{f_i}^*\left[f_i\left(t_i'\right)\right]}^{f_i^*(t_i') \text{ (Eq. 12)}}}_{\text{structured noise prediction}}, \qquad (6)$$

where $E^*$ denotes an expectation conditioned on $\vec{y}_i, X_i, \Theta$. In moving from Eq. 4 to 5, we have written the integral as an expectation and substituted the inner expectation with the mean of the normal distribution in Eq. 1. From Eq. 5 to 6, we use linearity of expectation. Eqs. 7, 10, and 12

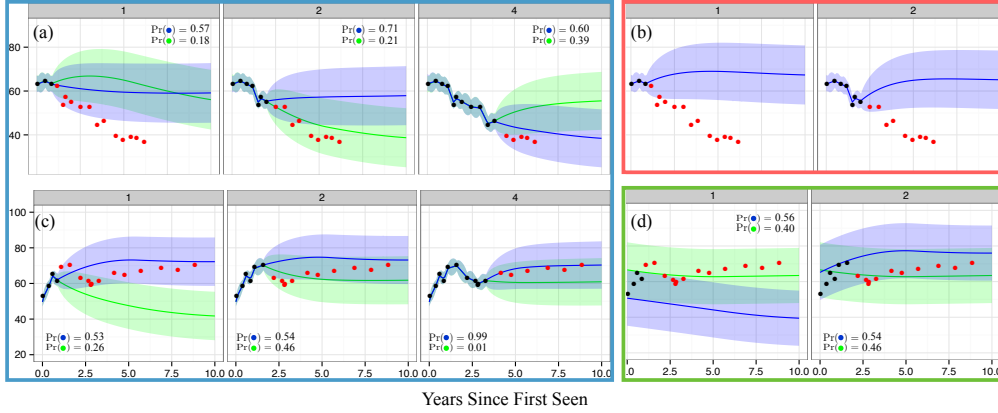

Figure 2: Plots (a) and (c) show dynamic predictions using the proposed model for two individuals. Red markers are unobserved. Blue shows the trajectory predicted using the most likely subtype, and green shows the second most likely. Plot (b) shows dynamic predictions using the B-spline GP baseline. Plot (d) shows predictions made using the proposed model without individual-specific adjustments.

below show how the expectations in Eq. 6 are computed. An expanded version of these steps are provided in the supplement.

Computing the population prediction is straightforward as all quantities are observed. To compute the subpopulation prediction, we need to compute the marginal posterior over $z_i$, which we used in the expectation step above (Eq. 3). The expected subtype coefficients are therefore

$$\vec{\beta}_i^* \triangleq \left( \sum_{z_i=1}^G \pi_{iz_i}^* \vec{\beta}_{z_i} \right). \tag{7}$$

To compute the individual prediction, note that by conditioning on $z_i$, the integral over the likelihood with respect to $f_i$ and the prior over $\vec{b}_i$ form the likelihood and prior of a Bayesian linear regression. Let $K_f = K_{\text{OU}}(\vec{t}_i, \vec{t}_i) + \sigma^2 \boldsymbol{I}$, then the posterior over $\vec{b}_i$ conditioned on $z_i$ is:

$$P\left( \vec{b}_i \mid z_i, \vec{y}_i, X_i, \Theta \right) \propto \mathcal{N}\left( \vec{b}_i \mid 0, \Sigma_b \right) \mathcal{N}\left( \vec{y}_i \mid \Phi_p \Lambda \, \vec{x}_{ip} + \Phi_z \left( \vec{t}_i \right) \vec{\beta}_{z_i} + \Phi_\ell \left( \vec{t}_i \right) \vec{b}_i, \, K_f \right). \tag{8}$$

Just as in Eq. 2, we have integrated over $f_i$ moving its effect from the mean of the normal distribution to the covariance. Because the prior over $\vec{b}_i$ is conjugate to the likelihood on the right side of Eq. 8, the posterior can be written in closed form as a normal distribution (see e.g. [10]). The mean of the left side of Eq. 8 is therefore

$$\left[ \Sigma_b^{-1} + \Phi_\ell(\vec{t}_i)^\top K_f^{-1} \Phi_\ell(\vec{t}_i) \right]^{-1} \left[ \Phi_\ell(\vec{t}_i)^\top K_f^{-1} \left( \vec{y}_i - \Phi_p(\vec{t}_i) \Lambda \, \vec{x}_{ip} - \Phi_z(\vec{t}_i) \vec{\beta}_{z_i} \right) \right], \tag{9}$$

To compute the unconditional posterior mean of $\vec{b}_i$ we take the expectation of Eq. 9 with respect to the posterior over $z_i$. Eq. 9 is linear in $\vec{\beta}_{z_i}$, so we can directly replace $\vec{\beta}_{z_i}$ with its mean (Eq. 7):

$$\vec{b}_i^* \triangleq \left[ \Sigma_b^{-1} + \Phi_\ell(\vec{t}_i)^\top K_f^{-1} \Phi_\ell(\vec{t}_i) \right]^{-1} \left[ \Phi_\ell(\vec{t}_i)^\top K_f^{-1} \left( \vec{y}_i - \Phi_p(\vec{t}_i) \Lambda \, \vec{x}_{ip} - \Phi_z(\vec{t}_i) \vec{\beta}_i^* \right) \right]. \tag{10}$$

Finally, to compute the structured noise prediction, note that conditioned on $z_i$ and $\vec{b}_i$, the GP prior and marker likelihood (Eq. 1) form a standard GP regression (see e.g. [20]). The conditional posterior of $f_i(t_i')$ is therefore a GP with mean

$$K_{\text{OU}}(t_i', \vec{t}_i) \left[ K_{\text{OU}}(\vec{t}_i, \vec{t}_i) + \sigma^2 \boldsymbol{I} \right]^{-1} \left( \vec{y}_i - \Phi_p(\vec{t}_i) \Lambda \, \vec{x}_{ip} - \Phi_z(\vec{t}_i) \vec{\beta}_{z_i} - \Phi_\ell(\vec{t}_i) \vec{b}_i \right). \tag{11}$$

To compute the unconditional posterior expectation of $f_i(t_i')$, we note that the expression above is linear in $z_i$ and $\vec{b}_i$ and so their expectations can be plugged in to obtain

$$f^*(t_i') \triangleq K_{\text{OU}}(t_i', \vec{t}_i) \left[ K_{\text{OU}}(\vec{t}_i, \vec{t}_i) + \sigma^2 \boldsymbol{I} \right]^{-1} \left( \vec{y}_i - \Phi_p(\vec{t}_i) \Lambda \, \vec{x}_{ip} - \Phi_z(\vec{t}_i) \vec{\beta}_i^* - \Phi_\ell(\vec{t}_i) \vec{b}_i^* \right). \tag{12}$$

# 3 Experiments

We demonstrate our approach by building a tool to predict the lung disease trajectories of individuals with scleroderma. Lung disease is currently the leading cause of death among scleroderma patients, and is notoriously difficult to treat because there are few predictors of decline and there is tremendous variability across individual trajectories [21]. Clinicians track lung severity using percent of predicted forced vital capacity (PFVC), which is expected to drop as the disease progresses. In addition, demographic variables and molecular test results are often available at baseline to aid prognoses. We train and validate our model using data from the Johns Hopkins Scleroderma Center patient registry, which is one of the largest in the world. To select individuals from the registry, we used the following criteria. First, we include individuals who were seen at the clinic within two years of their earliest scleroderma-related symptom. Second, we exclude all individuals with fewer than two PFVC measurements after their first visit. Finally, we exclude individuals who received a lung transplant. The dataset contains 672 individuals and a total of $4,992$ PFVC measurements.

For the population model, we use constant functions (i.e. observed covariates adjust an individual's intercept). The population covariates ($\vec{x}_{ip}$) are gender, African American race, and indicators of ACA and Scl-70 antibodies—two proteins believed to be connected to scleroderma-related lung disease. Note that all features are binary. For the subpopulation B-splines, we set boundary knots at 0 and 25 years (the maximum observation time in our data set is 23 years), use two interior knots that divide the time period from 0-25 years into three equally spaced chunks, and use quadratics as the piecewise components. These B-spline hyperparameters (knots and polynomial degree) are also used for all baseline models. We select $G = 9$ subtypes using BIC. The covariates in the subtype marginal model ($\vec{x}_{iz}$) are the same used in the population model. For the individual model, we use linear functions. For the hyper-parameters $\Theta_1 = \{\Sigma_b, \alpha, \ell, \sigma^2\}$ we set $\Sigma_b$ to be a diagonal covariance matrix with entries $[16, 10^{-2}]$ along the diagonal, which correspond to intercept and slope variances respectively. Finally, we set $\alpha = 6$, $\ell = 2$, and $\sigma^2 = 1$ using domain knowledge; we expect transient deviations to last around 2 years and to change PFVC by around $\pm 6$ units.

**Baselines.** First, to compare against typical approaches used in clinical medicine that condition on baseline covariates only (e.g. [22]), we fit a regression model conditioned on all covariates included in $\vec{x}_{iz}$ above. The mean is parameterized using B-spline bases ($\Phi(t)$) as:

$$\hat{y} \mid \vec{x}_{iz} = \Phi(t)^{\top} \left( \vec{\beta}_0 + \sum_{x_i \text{ in } \vec{x}_{iz}} x_i \vec{\beta}_i + \sum_{x_i, x_j \text{ in pairs of } \vec{x}_{iz}} x_i x_j \vec{\beta}_{ij} \right). \tag{13}$$

The second baseline is similar to [8] and [23] and extends the first baseline by accounting for individual-specific heterogeneity. The model has a mean function identical to the first baseline and individualizes predictions using a GP with the same kernel as in Equation 2 (using hyper-parameters as above). Another natural approach is to explain heterogeneity by using a mixture model similar to [9]. However, a mixture model cannot adequately explain away individual-specific sources of variability that are unrelated to subtype and therefore fails to recover subtypes that capture canonical trajectories (we discuss this in detail in the supplemental section). The recovered subtypes from the full model do not suffer from this issue. To make the comparison fair and to understand the extent to which the individual-specific component contributes towards personalizing predictions, we create a mixture model (Proposed w/ no personalization) where the subtypes are fixed to be the same as those in the full model and the remaining parameters are learned. Note that this version does not contain the individual-specific component.

**Evaluation.** We make predictions after one, two, and four years of follow-up. Errors are summarized within four disjoint time periods: $(1, 2]$, $(2, 4]$, $(4, 8]$, and $(8, 25]$ years[2]. To measure error, we use the absolute difference between the prediction and a smoothed version of the individual's observed trajectory. We estimate mean absolute error (MAE) using 10-fold CV at the level of individuals (i.e. all of an individual's data is held-out), and test for statistically significant reductions in error using a one-sided, paired t-test. For all models, we use the MAP estimate of the individual's trajectory. In the models that include subtypes, this means that we choose the trajectory predicted by the most likely subtype under the posterior. Although this discards information from the posterior, in our experience clinicians find this choice to be more interpretable.

**Qualitative results.** In Figure 2 we present dynamically updated predictions for two patients (one per row, dynamic updates move left to right). Blue lines indicate the prediction under the most likely subtype and green lines indicate the prediction under the second most likely. The first individual

| Predictions using 1 year of data | | | | | | | | |
|---|---|---|---|---|---|---|---|---|
| Model | (1, 2] | % Im. | (2, 4] | % Im. | (4, 8] | % Im. | (8, 25] | % Im. |
| B-spline with Baseline Feats. | 12.78 | | 12.73 | | 12.40 | | 12.14 | |
| B-spline + GP | 5.49 | | 7.70 | | **9.67** | | **10.71** | |
| Proposed | **5.26** | | *__7.04__ | 8.6 | 10.17 | | 12.12 | |
| Proposed w/ no personalization | 6.17 | | 7.12 | | 9.38 | | 12.85 | |
| Predictions using 2 years of data | | | | | | | | |
| B-spline with Baseline Feats. | | | 12.73 | | 12.40 | | 12.14 | |
| B-spline + GP | | | 5.88 | | 8.65 | | 10.02 | |
| Proposed | | | *__5.48__ | 6.8 | *__7.95__ | 8.1 | **9.53** | |
| Proposed w/ no personalization | | | 6.00 | | 8.12 | | 11.39 | |
| Predictions using 4 years of data | | | | | | | | |
| B-spline with Baseline Feats. | | | | | 12.40 | | 12.14 | |
| B-spline + GP | | | | | 6.00 | | 8.88 | |
| Proposed | | | | | *__5.14__ | 14.3 | *__7.58__ | 14.3 |
| Proposed w/ no personalization | | | | | 5.75 | | 9.16 | |

Table 1: MAE of PFVC predictions for the two baselines and the proposed model. Bold numbers indicate best performance across models (* is stat. significant). "% Im." reports percent improvement over next best.

(Figure 2a) is a 50-year-old, white woman with Scl-70 antibodies, which are thought to be associated with active lung disease. Within the first year, her disease seems stable, and the model predicts this course with 57% confidence. After another year of data, the model shifts 21% of its belief to a rapidly declining trajectory; likely in part due to the sudden dip in year 2. We contrast this with the behavior of the B-spline GP shown in Figure 2b, which has limited capacity to express individualized long-term behavior. We see that the model does not adequately adjust in light of the downward trend between years one and two. To illustrate the value of including individual-specific adjustments, we now turn to Figures 2c and 2d (which plot predictions made by the proposed model with and without personalization respectively). This individual is a 60-year-old, white man that is Scl-70 negative, which makes declining lung function less likely. Both models use the same set of subtypes, but whereas the model without individual-specific adjustment does not consider the recovering subtype to be likely until after year two, the full model shifts the recovering subtype trajectory downward towards the man's initial PFVC value and identify the correct trajectory using a single year of data.

**Quantitative results.** Table 1 reports MAE for the baselines and the proposed model. We note that after observing two or more years of data, our model's errors are smaller than the two baselines (and statistically significantly so in all but one comparison). Although the B-spline GP improves over the first baseline, these results suggest that both subpopulation and individual-specific components enable more accurate predictions of an individual's future course as more data are observed. Moreover, by comparing the proposed model with and without personalization, we see that subtypes alone are not sufficient and that individual-specific adjustments are critical. These improvements also have clinical significance. For example, individuals who drop by more than 10 PFVC are candidates for aggressive immunosuppressive therapy. Out of the 7.5% of individuals in our data who decline by more than 10 PFVC, our model predicts such a decline at twice the true-positive rate of the B-spline GP (31% vs. 17%) and with a lower false-positive rate (81% vs. 90%).

## 4 Conclusion

We have described a hierarchical model for making individualized predictions of disease activity trajectories that accounts for both latent and observed sources of heterogeneity. We empirically demonstrated that using all elements of the proposed hierarchy allows our model to dynamically personalize predictions and reduce error as more data about an individual is collected. Although our analysis focused on scleroderma, our approach is more broadly applicable to other complex, heterogeneous diseases [1]. Examples of such diseases include asthma [3], autism [4], and COPD [5]. There are several promising directions for further developing the ideas presented here. First, we observed that predictions are less accurate early in the disease course when little data is available to learn the individual-specific adjustments. To address this shortcoming, it may be possible to leverage time-dependent covariates in addition to the baseline covariates used here. Second, the quality of our predictions depends upon the allowed types of individual-specific adjustments encoded in the model. More sophisticated models of individual variation may further improve performance. Moreover, approaches for automatically learning the class of possible adjustments would make it possible to apply our approach to new diseases more quickly.

## Footnotes

[1]The model focuses on predicting the long-term trajectory of an individual when left untreated. In many chronic conditions, as is the case for scleroderma, drugs only provide short-term relief (accounted for in our model by the individual-specific adjustments). If treatments that alter long-term course are available and commonly prescribed, then these should be included within the model as an additional component that influences the trajectory.

[2]After the eighth year, data becomes too sparse to further divide this time span.

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
