[Supplementary Material]

# A Framework for Individualizing Predictions of Disease Trajectories by Exploiting Multi-Resolution Structure: Supplement

**Peter Schulam**
Dept. of Computer Science
Johns Hopkins University
Baltimore, MD 21218
pschulam@jhu.edu

**Suchi Saria**
Dept. of Computer Science
Johns Hopkins University
Baltimore, MD 21218
ssaria@cs.jhu.edu

## 1 Expectation Maximization for Disease Trajectory Model

### 1.1 Objective

We include the derivation of the EM objective for convenience. Recall that the model for marker $y_{ij}$ given parameters $\Theta$ and latent variables $\{z_i, \vec{b}_i, f_i\}$ is

$$y_{ij} \sim \mathcal{N}\left( \underbrace{\Phi_p(t_{ij})^\top \Lambda \; \vec{x}_{ip}}_{\text{(A) population}} + \underbrace{\Phi_z(t_{ij})^\top \vec{\beta}_{z_i}}_{\text{(B) subpopulation}} + \underbrace{\Phi_\ell(t_{ij})^\top \vec{b}_i}_{\text{(C) individual}} + \underbrace{f_i(t_{ij})}_{\text{(D) structured noise}}, \sigma^2 \right). \quad (1)$$

Let $X_i = \{\vec{t}_i, \vec{x}_{ip}, \vec{x}_{iz}\}$ denote individual $i$'s observation times and features, then marginalizing the joint likelihood gives us

$$P\left(\vec{y}_i \mid X_i, \Theta\right)$$

$$= \sum_{z_i=1}^{G} \underbrace{P\left(z_i \mid X_i, \Theta\right)}_{\text{Mult. regression prior}} \int_{\mathbb{R}^{d_\ell}} \underbrace{P\left(\vec{b}_i \mid \Theta\right)}_{\text{Normal prior}} \int_{\mathbb{R}^{N_i}} \underbrace{P\left(f_i \mid \Theta\right)}_{\text{GP prior}} \underbrace{P\left(\vec{y}_i \mid z_i, \vec{b}_i, f_i, X_i, \Theta\right)}_{\text{Eq. 1}} df_i \, d\vec{b}_i \quad (2)$$

$$= \sum_{z_i=1}^{G} \pi_{z_i}\left(\vec{x}_{iz}\right) \mathcal{N}\left(\vec{y}_i \mid \Phi_p\left(\vec{t}_i\right) \Lambda \vec{x}_{ip} + \Phi_z\left(\vec{t}_i\right) \vec{\beta}_{z_i}, K\left(\vec{t}_i, \vec{t}_i\right)\right). \quad (3)$$

Moving from Eq. 2 to Eq. 3, we evaluate the innermost integral using the fact that the GP prior over $f_i$ is conjugate to Eq. 1 yielding a new multivariate normal [1]. To evaluate the next integral in Eq. 2, we again have that the normal prior over $\vec{b}_i$ is conjugate to the multivariate normal obtained by marginalizing over $f_i$, which gives us the multivariate normal shown in Eq. 3 where the covariance function is defined as

$$K\left(t_1, t_2\right) = \Phi_\ell\left(t_1\right)^\top \Sigma_b \Phi_\ell\left(t_2\right) + K_{\text{OU}}\left(t_1, t_2\right) + \sigma^2 \mathbb{I}\left(t_1 = t_2\right). \quad (4)$$

We see that the observed-data log-likelihood for individual $i$ is defined by a mixture of multivariate normals where each subtype is associated with a class in the mixture. The mixing probabilities are defined by the multinomial logistic regression. The mean of the multivariate normal is defined by the population and subpopulation models, and the covariance is defined by the individual and structured noise models. The observed-data log-likelihood for all individuals is therefore

$$\mathcal{L}\left(\Theta\right) = \sum_{i=1}^{M} \log\left[\sum_{z_i=1}^{G} \pi_{z_i}\left(\vec{x}_{iz}\right) \mathcal{N}\left(\vec{y}_i \mid \Phi_p\left(\vec{t}_i\right) \Lambda \vec{x}_{ip} + \Phi_z\left(\vec{t}_i\right) \vec{\beta}_{z_i}, K\left(\vec{t}_i, \vec{t}_i\right)\right)\right]. \quad (5)$$

## 1.2 Expectation Step

All parameters related to $\vec{b}_i$ and $f_i$ are limited to the covariance kernel and are therefore not optimized using EM. We therefore only need to consider the mechanism indicators $z_i$ as unobserved in the expectation step. Because $z_i$ is discrete, its posterior is simply computed using the joint probability of $z_i$ and $\vec{y}_i$. Let $\pi_{ig}^*$ denote the posterior probability that individual $i$ has mechanism $g \in \{1, \ldots, G\}$, then we have

$$\pi_{ig}^* \propto \pi_g\left(\vec{x}_{iz}\right) \mathcal{N}\left(\vec{y}_i \mid \Phi_p\left(\vec{t}_i\right) \Lambda \, \vec{x}_{ip} + \Phi_z\left(\vec{t}_i\right) \vec{\beta}_g, K\left(\vec{t}_i, \vec{t}_i\right)\right). \tag{6}$$

## 1.3 Maximization Step

In the maximization step, we maximize the expected complete-data log likelihood with respect to $\Theta_2$. We can write the complete-data log likelihood as

$$\mathcal{L}_c\left(\Theta_2\right) = \sum_{i=1}^{M} \log \pi_{z_i}\left(\vec{x}_{iz}\right) + \log \mathcal{N}\left(\vec{y}_i \mid \Phi_p\left(\vec{t}_i\right) \Lambda \, \vec{x}_{ip} + \Phi_z\left(\vec{t}_i\right) \vec{\beta}_{z_i}, K\left(\vec{t}_i, \vec{t}_i\right)\right)$$

$$= \underbrace{\sum_{i=1}^{M} \log \pi_{z_i}\left(\vec{x}_{iz}\right)}_{\text{Subtype marginal}} + \underbrace{\sum_{i=1}^{M} \log \mathcal{N}\left(\vec{y}_i \mid \Phi_p\left(\vec{t}_i\right) \Lambda \, \vec{x}_{ip} + \Phi_z\left(\vec{t}_i\right) \vec{\beta}_{z_i}, K\left(\vec{t}_i, \vec{t}_i\right)\right)}_{\text{Marker conditional}}. \tag{7}$$

The expectation is taken with respect to the posterior over $z_i$.

To maximize this objective with respect to $\vec{w}_{1:G}$, we can focus on the first term in the sum above. By writing the log probabilities in full form we have

$$\sum_{i=1}^{M} \vec{w}_{z_i}^{\top} \vec{x}_{iz} - \log\left(\sum_{g'=1}^{G} e^{\vec{w}_{g'}^{\top} \vec{x}_i}\right) = \sum_{i=1}^{M} \sum_{g=1}^{G} \mathbb{I}\left(z_i = g\right)\left[\vec{w}_{z_i}^{\top} \vec{x}_{iz} - \log\left(\sum_{g'=1}^{G} e^{\vec{w}_{g'}^{\top} \vec{x}_i}\right)\right]. \tag{8}$$

Taking the expectation of this expression with respect to the posterior over $z_i$ amounts to replacing the indicator function $\mathbb{I}\left(z_i = g\right)$ with the posterior probability $\pi_{ig}^*$. We can maximize this expression with respect to the multinomial logistic regression parameters $\vec{w}_{1:G}$ using gradient-based methods.

The second term in the complete-data log-likelihood involves the population feature-coefficient map $\Lambda$ and the subpopulation model coefficients $\vec{\beta}_{1:G}$. These two sets of parameters are coupled in the exponential term of the multivariate normal, and therefore must be optimized jointly. To optimize these parameters, we first note that the multivariate normal log-likelihood can be rewritten as a weighted least squares problem. To see this, we first write out the second term in Equation 7 using the log of the multivariate normal density.

$$\sum_{i=1}^{M} -\frac{1}{2}\left(\vec{y}_i - \Phi_p\left(\vec{t}_i\right) \Lambda \, \vec{x}_{ip} - \Phi_z\left(\vec{t}_i\right) \vec{\beta}_{z_i}\right)^{\top} K\left(\vec{t}_i, \vec{t}_i\right)^{-1}\left(\vec{y}_i - \Phi_p\left(\vec{t}_i\right) \Lambda \, \vec{x}_{ip} - \Phi_z\left(\vec{t}_i\right) \vec{\beta}_{z_i}\right) + C_i,$$
$$\tag{9}$$

where the constant $C_i$ is the log normalizing constant, which does not contain the parameters of interest. Maximizing this expression with respective $\vec{\beta}_g$ for each $g \in \{1, \ldots, G\}$ and $\Lambda$ is equivalent to minimizing the negative value of the quadratic. Let $W_i = K(\vec{t}_i, \vec{t}_i)^{-1}$, then we can write the negative of the quadratic term as a weighted least squares objective with weight matrix $W_i$. Given $\Lambda$, we can optimize $\vec{\beta}_{1:G}$ in closed form using the standard weighted normal equations. Similarly, given $\vec{\beta}_{1:G}$, we can optimize $\Lambda$. This suggests an alternating strategy wherein we iteratively refine the subtype parameters given the control parameters and vice versa.

Given a current estimate of $\Lambda$, the sufficient statistics needed to optimize $\beta_g$ are computed for each individual $i$ belonging to subtype $g$. These statistics are

$$\eta_g^{(i1)} = \Phi_z^{\top}\left(\vec{t}_i\right) W_i \, \Phi_z\left(\vec{t}_i\right), \tag{10}$$

$$\eta_g^{(i2)} = \Phi_z^{\top}\left(\vec{t}_i\right) W_i \left(\vec{y}_i - \Phi_p\left(\vec{t}_i\right) \Lambda \, \vec{x}_{ip}\right). \tag{11}$$

To maximize $\vec{\beta}_g$ we leverage two facts from statistics. First, the sufficient statistics required to compute the maximum likelihood estimate from $M$ independent weighted linear regressions is simply the sum of the individual sufficient statistics. Second, when maximizing an expected complete-data log-likelihood, we can replace the sufficient statistics with expected sufficient statistics. In this case, we multiply $\eta_g^{(i1)}$ and $\eta_g^{(i2)}$ by the posterior probability over $\mathbb{I}(z_i = g)$. We can therefore compute the optimal value for $\vec{\beta}_g$ using

$$\vec{\beta}_g = \left( \sum_{i=1}^{M} \pi_{ig}^* \eta_g^{(i1)} \right)^{-1} \left( \sum_{i=1}^{M} \pi_{ig}^* \eta_g^{(i2)} \right). \tag{12}$$

Similarly, given a current estimate of $\vec{\beta}_{1:G}$, the sufficient statistics needed to optimize $\Lambda$ are computed across all individuals. Let $\vec{\Lambda}$ denote the vectorization of the feature-coefficient map matrix (i.e. the column vector obtained by stacking the columns of $\Lambda$) and let $\Phi_p^{(\vec{x}_{ip})}\left(\vec{t}_i\right) = \left[ \Phi_p\left(\vec{t}_i\right) x_{ip,1}, \ldots, \Phi_p\left(\vec{t}_i\right) x_{ip,q_p} \right]$, then the sufficient statistics for optimizing $\vec{\Lambda}$ are

$$\eta_\Lambda^{(i1)} = \Phi_p^{(\vec{x}_{ip})}\left(\vec{t}_i\right)^\top W_i \, \Phi_p^{(\vec{x}_{ip})}\left(\vec{t}_i\right), \tag{13}$$

$$\eta_\Lambda^{(i2)} = \Phi_p^{(\vec{x}_{ip})}\left(\vec{t}_i\right)^\top W_i \left(\vec{y}_i - \Phi_z\left(\vec{t}_i\right) \mathbb{E}\left[\beta_{z_i}\right]\right), \tag{14}$$

where the expectation in the second sufficient statistic is taken with respect to the posterior over $z_i$. Given these sufficient statistics, the optimal value for $\vec{\Lambda}$ is

$$\vec{\Lambda} = \left( \sum_{i=1}^{M} \eta_\Lambda^{(i1)} \right)^{-1} \left( \sum_{i=1}^{M} \eta_\Lambda^{(i2)} \right). \tag{15}$$

## 2   Prediction

Our prediction $\hat{y}(t_i')$ for the value of the trajectory at time $t_i'$ is the expectation of the marker $y_i'$ under the posterior predictive conditioned on observed markers $\vec{y}_i$ measured at times $\vec{t}_i$ thus far. This requires evaluating the following expression:

$$\hat{y}\left(t_i'\right) = \sum_{z_i=1}^{G} \int_{R^{d_\ell}} \int_{R^{N_i}} \underbrace{\mathbb{E}\left[y_i' \mid z_i, \vec{b}_i, f_i, t_i'\right]}_{\text{prediction given latent vars.}} \underbrace{P\left(z_i, \vec{b}_i, f_i \mid \vec{y}_i, X_i, \Theta\right)}_{\text{posterior over latent vars.}} df_i \, d\vec{b}_i \tag{16}$$

$$= \mathbb{E}_{z_i, \vec{b}_i, f_i}^* \left[ \Phi_p\left(t_i'\right)^\top \Lambda \, \vec{x}_{ip} + \Phi_z\left(t_i'\right)^\top \vec{\beta}_{z_i} + \Phi_\ell\left(t_i'\right)^\top \vec{b}_i + f_i\left(t_i'\right) \right] \tag{17}$$

$$= \underbrace{\Phi_p\left(t_i'\right)^\top \Lambda \, \vec{x}_{ip}}_{\text{population prediction}} + \Phi_z\left(t_i'\right)^\top \overbrace{\mathbb{E}_{z_i}^*\left[\vec{\beta}_{z_i}\right]}^{\vec{\beta}_i^* \text{ (Eq. 19)}} + \Phi_\ell\left(t_i'\right)^\top \overbrace{\mathbb{E}_{\vec{b}_i}^*\left[\vec{b}_i\right]}^{\vec{b}_i^* \text{ (Eq. 23)}} + \overbrace{\mathbb{E}_{f_i}^*\left[f_i\left(t_i'\right)\right]}^{f_i^*(t_i') \text{ (Eq. 30)}}, \tag{18}$$

$$\underbrace{\phantom{\Phi_p\left(t_i'\right)^\top \Lambda \, \vec{x}_{ip}}}_{\text{population prediction}} \quad \underbrace{\phantom{\Phi_z\left(t_i'\right)^\top \mathbb{E}_{z_i}^*\left[\vec{\beta}_{z_i}\right]}}_{\text{subpopulation prediction}} \quad \underbrace{\phantom{\Phi_\ell\left(t_i'\right)^\top \mathbb{E}_{\vec{b}_i}^*\left[\vec{b}_i\right]}}_{\text{individual prediction}} \quad \underbrace{\phantom{\mathbb{E}_{f_i}^*\left[f_i\left(t_i'\right)\right]}}_{\text{structured noise prediction}}$$

where $E^*$ denotes an expectation conditioned on $\vec{y}_i, X_i, \Theta$. In moving from Eq. 16 to 17, we have written the integral as an expectation and substituted the inner expectation with the mean of the normal distribution in Eq. 1. From Eq. 17 to 18, we use linearity of expectation. Eqs. 19, 23, and 30 below show how the expectations in Eq. 18 are computed.

Computing the population prediction is straightforward as all quantities are observed. To compute the subpopulation prediction, we need to compute the marginal posterior over $z_i$, which we used in the expectation step above (Eq. 6). The expected subtype coefficients are therefore

$$\vec{\beta}_i^* \triangleq \left( \sum_{z_i=1}^{G} \pi_{iz_i}^* \vec{\beta}_{z_i} \right). \tag{19}$$

To compute the individual prediction, note that by conditioning on $z_i$ and integrating over $f_i$, the innermost integral from Eq. 2 and the prior over $\vec{b}_i$ form the likelihood and prior of a Bayesian linear regression. Let $K_f = K_{\text{OU}}(\vec{t}_i, \vec{t}_i) + \sigma^2 I$, then the posterior over $\vec{b}_i$ conditioned on $z_i$ is:

$$P\left(\vec{b}_i \mid z_i, \vec{y}_i, X_i, \Theta\right) \propto \mathcal{N}\left(\vec{b}_i \mid 0, \Sigma_b\right) \mathcal{N}\left(\vec{y}_i \mid \Phi_p \Lambda \, \vec{x}_{ip} + \Phi_z\left(\vec{t}_i\right) \vec{\beta}_{z_i} + \Phi_\ell\left(\vec{t}_i\right) \vec{b}_i, K_f\right). \tag{20}$$

Just as in Eq. 3, we have integrated over $f_i$ moving its effect from the mean of the normal distribution to the covariance. Because the prior over $\vec{b}_i$ is conjugate to the likelihood on the right side of Eq. 20, the posterior can be written in closed form as a normal distribution with the following mean and variance (see e.g. [2]).

$$\Sigma_b^* = \left[\Sigma_b^{-1} + \Phi_\ell(\vec{t_i})^\top K_f^{-1} \Phi_\ell(\vec{t_i})\right]^{-1}, \tag{21}$$

$$\mu_b^* = \Sigma_b^* \left[\Phi_\ell(\vec{t_i})^\top K_f^{-1} \left(\vec{y}_i - \Phi_p(\vec{t_i})\Lambda\,\vec{x}_{ip} - \Phi_z(\vec{t_i})\vec{\beta}_{z_i}\right)\right]. \tag{22}$$

We are interested in the posterior expectation of this normal distribution, but have conditioned on $z_i$. We can derive the unconditional posterior mean of $\vec{b}_i$ by computing the expectation of $\mu_b^*$ with respect to the posterior over $z_i$ (Eq. 6). The only term involving $z_i$ in $\mu_b^*$ is $\vec{\beta}_{z_i}$. Furthermore, $\mu_b^*$ is linear in $\vec{\beta}_{z_i}$ so we can simply replace $\vec{\beta}_{z_i}$ with its expectation under the posterior, which we've already computed in Eq. 19. This gives us

$$\vec{b}_i^* \triangleq \left[\Sigma_b^{-1} + \Phi_\ell(\vec{t_i})^\top K_f^{-1} \Phi_\ell(\vec{t_i})\right]^{-1} \left[\Phi_\ell(\vec{t_i})^\top K_f^{-1} \left(\vec{y}_i - \Phi_p(\vec{t_i})\Lambda\,\vec{x}_{ip} - \Phi_z(\vec{t_i})\vec{\beta}_i^*\right)\right]. \tag{23}$$

Finally, to compute the structured noise prediction, recall from Equation 2 that when conditioned on $z_i$ and $\vec{b}_i$ the GP prior and marker likelihood (Eq. 1) form a standard GP regression (see e.g. [1]). To see this, note that by conditioning on $z_i$ and $\vec{b}_i$ we can compute the residuals of the observed marker values:

$$\vec{r}_i = \left(\vec{y}_i - \Phi_p(\vec{t_i})\Lambda\,\vec{x}_{ip} - \Phi_z(\vec{t_i})\vec{\beta}_{z_i} - \Phi_\ell(\vec{t_i})\vec{b}_i\right). \tag{24}$$

To explain the remaining variation we use $f_i(\vec{t_i})$, which we know has a Gaussian process prior with OU covariance kernel. Moreover, given $f_i(\vec{t_i})$ the residuals $\vec{r}_i$ are normally distributed.

$$\vec{r}_i \sim \mathcal{N}\left(f_i(\vec{t_i}), \sigma^2 \boldsymbol{I}\right). \tag{25}$$

To compute the value of the latent function at a new point $t_i'$, we use the posterior predictive

$$P\left(f_i(t_i') \mid \vec{r}_i\right) = \int_{R^{N_i}} P\left(f_i(t_i') \mid f_i(\vec{t_i})\right) P\left(f_i(\vec{t_i}) \mid \vec{r}_i\right) df_i(\vec{t_i}), \tag{26}$$

where

$$P\left(f_i(\vec{t_i}) \mid \vec{r}_i\right) \propto \mathcal{N}\left(\vec{r}_i \mid f_i(\vec{t_i}), \sigma^2 \boldsymbol{I}\right) \mathrm{GP}\left(f(\vec{t_i}) \mid 0, K_{\mathrm{OU}}(\vec{t_i}, \vec{t_i})\right). \tag{27}$$

The posterior predictive is itself a GP. The mean of this GP is used to predict new observations in GP regression [1]. Using standard results from [1] we have that the conditional expectation of $f_i(t_i')$ given $z_i$ and $\vec{b}_i$ is

$$K_{\mathrm{OU}}(t_i', \vec{t_i}) \left[K_{\mathrm{OU}}(\vec{t_i}, \vec{t_i}) + \sigma^2 \boldsymbol{I}\right]^{-1} \vec{r}_i = \tag{28}$$

$$K_{\mathrm{OU}}(t_i', \vec{t_i}) \left[K_{\mathrm{OU}}(\vec{t_i}, \vec{t_i}) + \sigma^2 \boldsymbol{I}\right]^{-1} \left(\vec{y}_i - \Phi_p(\vec{t_i})\Lambda\,\vec{x}_{ip} - \Phi_z(\vec{t_i})\vec{\beta}_{z_i} - \Phi_\ell(\vec{t_i})\vec{b}_i\right). \tag{29}$$

Just as with $\vec{b}_i$, we need to take the expectation of this expression with respect to the posteriors over $z_i$ and $\vec{b}_i$ to obtain the unconditional posterior expectation. This is easy to do because the expression above is linear in both $\vec{\beta}_{z_i}$ and $\vec{b}_i$ so we can simply replace them with their expectations computed in Eq. 19 and Eq. 23 respectively:

$$f^*(t_i') \triangleq K_{\mathrm{OU}}(t_i', \vec{t_i}) \left[K_{\mathrm{OU}}(\vec{t_i}, \vec{t_i}) + \sigma^2 \boldsymbol{I}\right]^{-1} \left(\vec{y}_i - \Phi_p(\vec{t_i})\Lambda\,\vec{x}_{ip} - \Phi_z(\vec{t_i})\vec{\beta}_i^* - \Phi_\ell(\vec{t_i})\vec{b}_i^*\right). \tag{30}$$

## 3   Will a mixture model suffice?

A natural question one might ask is whether a B-spline mixture model would sufficiently explain the variability across individuals. In other words, is it necessary to include the individual-specific components that adjust for long-term and short-term deviations from the subtype trajectory in the proposed model? We fit a B-spline mixture model that is similar to the proposed model but that

Figure 1: Comparison of sample subtypes learned by (a) a B-spline mixture model without a personalization component, and (b) the proposed model.

Figure 2: In panel (a), we show examples of individuals with trajectories that do not fit any of the subtypes learned by the mixture model. In panel (b), we show the first (blue) and second (green) most likely subtypes assigned by the mixture model to two individuals with a recovering trajectory (top row) and two individuals with a rapidly declining trajectory (bottom row). Finally, in panel (c), we show the most likely subtypes assigned by the proposed model to the same individuals. For all predictions in panels (b) and (c), the individuals were not included in the training set of the model that was used.

does not account for individual-specific long-term and short-term components. This is also similar to the approach by Proust-Lima et al. [3], a commonly used technique that accounts for population heterogeneity using a mixture model. In Figures 1a and 1b, we plot the subtypes learned from a random fold of our data using the B-spline mixture model and the proposed model respectively. We used BIC to determine the number of subtypes used in the B-spline mixture model. We find that a model that does not account for individual-specific variability is unable to recover *clinically-salient* subtypes that capture the different kinds of trajectories clinicians expect to see. In particular, Figure 1b highlights in red two types of trajectories that the B-spline mixture model (Figure 1a) is unable to learn: a rapidly declining subtype and recovering subtype.

In Figure 2a, we show data from several example individuals that fall under these subtypes. Figures 2b and 2c show the most likely (blue) and second most likely (green) subtypes assigned by the B-spline mixture model and the proposed model on two example patients from each of the two groups

shown in 2a. We note that in all four cases, there are no suitable subtypes in the B-spline mixture model. On the other hand, the proposed model recovers subtypes that generalize well and are able to capture the trajectories of the individuals shown. The behavior of the B-spline mixture model is not surprising because it has limited means for explaining away individual-specific long-term and short-term deviations from the subtype. This issue has also been discussed by Schulam et al. [4] in the context of subtype discovery. It is worth noting that since BIC was used for model selection, simply increasing the number of subtypes in the B-spline mixture model would not address this issue.