[Reviews · NeurIPS 2015]

Submitted by Assigned_Reviewer_1

The authors design and fit a hierarchical Bayesian model for predicting disease trajectories (i.e., a scalar measure of disease severity measured throughout the course of the disease) for individual patients. The overall model is an additive combination of a a number of terms including: (1) a population-level term, (2) a subpopulation term, (3) an individual term, (4) a GP term for structured errors. Each of these terms is a function of time, which is modeled parametrically in terms of the coefficients on pre-defined basis expansions (linear and/or B-splines). The subpopulation term involves a discrete mixture model, and the individual level term is a Bayesian linear regression. Distributions are chosen to be Gaussian, which makes most steps of inference and learning work out nicely. The discrete mixture leads to a GMM-like EM algorithm. The authors tune hyperparameters by hand, use likelihood-based learning (EM) for regression coefficients, and use posterior inference to predict the disease trajectory as it evolves. The qualitative results suggest that the method is effective at picking up the signal in lung-disease trajectories to make accurate predictions. The quantitative results demonstrate significant improvements over reasonable baseline.

Quality:

I find this to be a well done application paper. The model is not earth shattering, but not trivial either. It is clearly explained and appropriate to the problem at hand. The inference and learning methods are carefully derived. The experiments are designed well and demonstrate the value of the method.

Note: for the qualitative results, please comment on the range of different results. Are the examples in Figure 2 typical, or were these chosen as particularly compelling examples?

Clarity:

See above. The paper is well written, and the methods are explained clearly. One small comment: the related work section was hard to understand without having first seeing some details of the proposed method---e.g., to understand why plain GPs would not do the job.

Originality:

To the best of my knowledge, the model structure proposed here is original. It reuses fairly standard components, so it is not highly original. Because most of the components of the model are Gaussian, the math generally works out nicely. But there is enough complexity in the model components (splines, discrete mixture model, Bayesian linear regression, GP) that the careful derivation of inference and learning updates represent a contribution. When that is all done, standard algorithms can solve the inference and learning problems.

Significance:

The primary contributions are in the application domain area (health modeling): the design of a new model, derivation of inference and learning algorithms, and demonstration that the method works on a real task. The model could potentially apply elsewhere, but there is not a convincing case made here of significance/relevance as a general-purpose model.
Summary: This is a solid application paper with a well-designed and interesting but not earth-shattering model. It is carefully executed, seems correct, and provides nice results on a real health application.

Submitted by Assigned_Reviewer_2

The paper presents a graphical model for predicting individual disease trajectories which uses population. subpopulation and individual models. This is a really important area of application for machine learning methods. The paper proposes and develops a graphical model for this approach. While the model itself relies on existing pieces, it is put together in a thoughtful way and the choices are clearly explained and well motivated. An EM algorithm is developed, and while this again is done in a straightforward manner, the derivations are technically correct. The empirical evaluation uses a real patient data set and offers both qualitative and quantitative results. The model outperforms well chosen competitors. The paper is very well written and the work presented, while fairly standard in terms of the theoretical development, is novel in terms of the application. While certain parts of the model are targeted specifically to scleroderma, the general approach is likely to inspire others to tackle other diseases using similar approaches (but considering subpopulation and individual models suited to the disease at hand). Overall this is a good contribution and one likely to have practical significance.
Summary: This is a good application paper, which addresses an important problem in a technically sound way and obtains good results on real data.

Submitted by Assigned_Reviewer_3

The paper proposes using a four part model that captures population, subpopulation, and indiviudal characteristics in addition to noise. The paper focuses on lung disease in patients with scleroderma.

The model is intuitive and well explained. The training/prediction parts are relatively straightforward.

The results on the paper seem promising and I appreciate how the authors as included a discussion about what this finding might mean in practice (i.e., how they could be used to take action).

Some other related work that may be of interest: http://cs.nyu.edu/~dsontag/papers/WanSonWan_kdd14.pdf http://wan.poly.edu/KDD2012/docs/p1095.pdf

While the paper does include a number of reasonable baselines, that control for the elements added to the model. However, you are left a bit wondering if any of the related work approaches could be applied to this problem and how they would do.

While this paper focuses on one specific disease as a case study, which is fine, I think that the paper would benefit from a bit of discussion about how general the method is. That is, a brief discussion about how this could be applied to other diseases.

A step further would be if it is applicable to other tasks outside of medicine (though this is less important as disease progression is clearly a significant application).
Summary: The paper proposes a four part model for disease progression and includes an interesting case study.

Submitted by Assigned_Reviewer_4

This paper appears to be a minor modification to previously published work on using graphical models to predict disease trajectories (see Schulam et al, AAAI 2015). The authors do not cite this effort or compare to it experimentally. A comparison of the proposed approach to this previous use of graphical models for disease trajectories would have been far more meaningful and informative than comparisons to simpler baselines.

Some more details about the study would also help. For example, how many patients were in the database at the start (and what % were removed)? Were there any biases introduced by removing patients who died, didn't have enough monitoring data, or got transplants? Is the mean average error achieved for PFVC meaningful? Are there other cases that are interesting beyond the two anecdotally described?
Summary: Limited novelty relative to recent advances.

Author Feedback
Author rebuttal: Overall, few complaints were made. We address the two major comments via General Response 1 and General Response 2 and the remainder thereafter.

General Response 1
Reviewers #1 and #4 suggested a discussion of applicability to other diseases would be beneficial.

The proposed work is applicable to many complex diseases [1] including COPD [2], multiple sclerosis [3], Crohn's [4], asthma [5] and autism [6]. These are diseases where clinical scales of severity are used to track activity and where heterogeneity in disease presentation---due to observed and unobserved factors similar to ours---has been discussed extensively. Thank you for pointing this out; a longer discussion will be added to the paper.

1] http://goo.gl/WLFRXe
2] http://goo.gl/Kuf5kR
3] http://goo.gl/AtsH9c
4] http://goo.gl/Hzgdw1
5] http://goo.gl/NGuOYZ
6] http://goo.gl/vCkfNr

General Response 2
Reviewer #1 remarked that while the proposed baselines are reasonable, can other approaches (Wang et al. 2014 and Zhou et al 2012) be applied? Reviewer #2 remarked about the appropriateness of the baselines used and relationship to Schulam et al., 2015.

Works by Rizopoulos (B-spline + GP baseline) and Proust-Lima et al (B-spline Mixture baseline) are state-of-the-art statistical models for dynamical prediction of disease trajectories in heterogeneous populations; recent applications have used these to predict disease trajectories in cancer [1], HIV [2] and multiple sclerosis [3].

1] http://goo.gl/7cO8SC
2] http://goo.gl/Ir3436
3] http://goo.gl/yxBqFv

Wang et al., 2014: They learn a generative model for how different secondary complications (comorbidities) are expressed as a function of the disease-stage. Their task is different from ours in many respects. First, they focus on modeling comorbidities and not continuous clinical markers. Second, they focus on retrospective analysis of population-level parameters rather than individual-specific prediction. They also do not explicitly capture subpopulation structure.

Zhou et al., 2012: This paper more closely matches our task but there are two key differences. First, in their setting, predictions use information only at baseline and do not include new data over time. Second, their study design assumes predictions are made only at fixed time points in the future; our model, as is required by our problem, updates continuously and makes predictions at arbitrary points in continuous time.

Schulam et al., 2015 (S15): S15, like Wang's, falls in the general class of solutions to DPM. Our work, like S15, models heterogeneity across individuals via subtypes. S15 also accounts for individual-specific differences but these parameters are treated as nuisance and marginalized as only population level parameters are of interest. A key difference between S15 and the present work is a shift in focus from retrospective analysis (where only population-level parameters are estimated offline) to individualized prediction where individual-level parameters must be estimated dynamically. Further, in the prediction setting, we explicitly learn conditional priors to improve estimation of the individual-specific differences, e.g. P(subtype |baseline covariates) is explicitly modeled using a multinomial logistic regression which introduces individualized priors over subtypes. Similarly, P(b_i |baseline covariates) can be modeled to improve estimates of individual regression parameters. These priors are combined with observed data dynamically to produce a posterior distribution over the full-trajectory.

Broadly, different from state-of-the-art works on disease trajectory prediction (e.g. Proust-Lima, Rizopoulos, Zhou), our work proposes a principled Bayesian framework for making individual-specific predictions by building a comprehensive model of heterogeneity at varying granularities (population, subpopulation, individual) while explaining away noise due to unobserved external factors.

Reviewer #2
1. 1200 patients were seen within 2 years of disease onset. Of these, 16 had a lung transplant, 350 had missing antibodies, and 150 had only one PFVC record. Transplant recipients are rare; removal should not introduce bias. Antibody and PFVC measurements are routinely recorded, and so missingness is uncorrelated with outcome.

Reviewer #4
1. The examples in Fig 2a are illustrative of the model's performance on other individuals in addition to those used as anecdotes. Fig 2b,2c are representative examples of common errors when comparing predictions between the baselines and our model.
2. Under Obama's Precision Medicine initiative, there is a call for new methods for individualized prognosis of disease. We believe health modeling is a significant area where the machine learning community can play an important role much like it has done in personalized recommendations and web search.

Some references were omitted to preserve author identity and will be included in the revised paper.